# A major trade-off between growth and defense in *Arabidopsis thaliana* can vanish in field conditions

Derek S. Lundberg [1,2]*, Sonja Kersten[1¤a], Ezgi Mehmetoğlu Boz[2],
Pratchaya Pramoj Na Ayutthaya[1¤b], Wangsheng Zhu[1,3], Karin Poersch[1], Wei Yuan[1],
Sophia Swartz[1¤c], David Müller[1], Ilja Bezrukov[1], HARVEST TEAM[¶], Detlef Weigel[1,4]*

**1** Department of Molecular Biology, Max Planck Institute for Biology Tübingen, Tübingen, Germany,
**2** Department of Plant Biology, Swedish University of Agricultural Sciences, Uppsala, Sweden, **3** State Key Laboratory of Maize Bio-breeding/College of Plant Protection, China Agricultural University, Beijing, China, **4** Institute for Bioinformatics and Medical Informatics, University of Tübingen, Tübingen, Germany

¤a Current address: Department of Livestock Population Genomics, Institute of Animal Science, University of Hohenheim, Stuttgart, Germany
¤b Current address: Department of Plant Microbe Interactions, Max Planck Institute for Plant Breeding Research, Cologne, Germany
¤c Current address: Innovative Genomics Institute, UC Berkeley, Berkeley, California, USA
¶ Membership of the HARVEST TEAM is provided in the Acknowledgments.
* derek.lundberg@slu.se (DSL); weigel@tue.mpg.de (DW)

## Abstract

When wild plants defend themselves from pathogens, this often comes with a trade-off: the same genes that protect a plant from disease can also reduce its growth and fecundity in the absence of pathogens. One protein implicated in a major growth-defense trade-off is ACCELERATED CELL DEATH 6 (ACD6), an ion channel that modulates salicylic acid (SA) synthesis to potentiate a wide range of defenses. Wild *Arabidopsis thaliana* populations maintain significant functional variation at the *ACD6* locus, with some alleles making the protein hyperactive. In the greenhouse, plants with hyperactive *ACD6* alleles are resistant to diverse pathogens, yet they are of smaller stature, their leaves senesce earlier, and they set fewer seeds compared to plants with the standard allele. We hypothesized that *ACD6* hyperactivity would not only affect the growth of microbial pathogens but also more generally change leaf microbiome assembly. To test this in an ecologically meaningful context, we compared plants with hyperactive, standard, and defective *ACD6* alleles in the same field-collected soil, both outdoors and in naturally lit and climate-controlled indoor conditions, taking advantage of near-isogenic lines as well as a natural accession and a CRISPR-edited derivative. We surveyed visual phenotypes, gene expression, hormone levels, seed production, and the microbiome in each environment. The genetic precision of CRISPR-edited plants allowed us to conclude that *ACD6* genotype had no effect on mature field plants in our setting, despite reproducibly dramatic effects on greenhouse plants. We conclude that additional abiotic and/or microbial

**Data availability statement:** All sequencing data have been deposited in the European Nucleotide Archive (ENA, https://www.ebi. ac.uk/ena). They can be accessed under project number PRJEB78910. Other data files and numerical data behind the figures can be found at: https://doi.org/10.5281/zenodo.15527338.

**Funding:** The work has been supported by a Human Frontiers Science Program (HFSP) Long-Term Fellowship to DL (https://www. hfsp.org/, LT000565/2015-L), a Wallenberg Academy Fellows grant to DL (https://kaw. wallenberg.org/en, 2021.0102), an ERC Synergy Grant (https://erc.europa.eu/homep- age, PATHOCOM 951444) to DW and the Max Planck Society (https://www.mpg.de/en) for DW. Funders did not play any role in study design, data collection and analysis, decision to publish, or preparation of the manuscript.

**Competing interests:** I have read the journal's policy and the authors of this manuscript have the following competing interests: DW holds equity in Computomics, which advises plant breeders. DW also consults for KWS SE, a globally active plant breeder and seed producer. The other authors have declared that no competing interests exist.

**Abbreviations:** ACD6, ACCELERATED CELL DEATH 6; ASVs, amplicon sequence variants; GO, Gene Ontology; HIF, heterogeneous inbred family; IAA, indole-3-acetic acid; IQR, interquartile range; ITS, internal transcribed spacer; JA, jasmonic-acid; PCA, principal component analysis; SA, salicylic acid; SAG, SA-2-O-β-D-glucoside.

signals present outdoors—but not in the greenhouse—greatly modulate *ACD6* activity. This raises the possibility that the fitness costs of other commonly studied immune system genes may be grossly misjudged without field studies.

## Introduction

Because organisms live in environments that are constantly changing, trade-offs are a key component of evolution. In plants, a prominent trade-off is that between growth and defense, and many experimental and observational results have been framed in this context [1–8]. One such example comes from the *ACCELERATED CELL DEATH 6* (*ACD6*) gene in *Arabidopsis thaliana*. ACD6 associates with other defense proteins including the plant surface receptor FLS2, which detects bacterial flagellin [9–11], and it acts as an ion channel for calcium [11]. *ACD6* expression, protein accumulation and activity are modulated by the plant defense hormone salicylic acid (SA), and ACD6, in turn, regulates SA levels in a positive feedback loop [9].

Similar to many other immune genes, there is substantial structural and sequence diversity in the *ACD6* genomic region across accessions of *A. thaliana* [12–15]. These have a wide range of activities, either on their own or as heteroallelic combinations, and their activity can be further modulated by second-site genetic variants [11–15]. The class of alleles first found in the accession Est-1 (*ACD6*-Est-1) appears hyperactive when compared to the standard allele in the reference accession Col-0 [12]. Relative to standard alleles, *ACD6*-Est-1 increases SA accumulation and *PR1* expression, enhances resistance to bacterial, fungal, oomycete pathogens, and likely even sucking insects [12]. These defense benefits are, however, counterbalanced by slower leaf initiation rate, smaller final size, late-onset necrosis in fully expanded leaves, and lower fecundity [12]. Thus, the *ACD6*-Est-1 allele provides both a blessing and a curse, underlying what can be considered a typical growth-defense trade-off [12,16]. This allele is nevertheless rather common in natural populations, with about 10%–20% of wild accessions carrying *ACD6*-Est-1-type alleles [11,12], consistent with *ACD6*-Est-1 conferring tangible fitness benefits in nature and balancing selection maintaining the allele despite a growth penalty.

The *ACD6*-Est-1 phenotype shares similarities with the one caused by the ethyl methanesulfonate (EMS)-induced gain-of-function mutation *acd6-1*, which also links stunted growth with enhanced immunity [9,10,17,18]. However, *acd6-1* phenotypes are greatly exaggerated under lower temperatures [11,13], which is typical for many autoimmune genotypes [19], making it unlikely that they will survive outside under the conditions that are typical for much of the natural range of *A. thaliana* [20]. Similar phenotypes as in plants with an *ACD6*-Est-1-type allele are produced by certain heteroallelic combinations at *ACD6*, with the difference that *ACD6* compound heterozygotes, but not *ACD6*-Est-1-dependent necrosis, can be suppressed in the greenhouse by elevated temperature [11,13,14]. Counterintuitively, such heterozygotes can be found in the wild, where temperatures during the typical *A. thaliana* growing season are generally lower than the ones used in laboratory experiments [13]. In addition, when grown outdoors, the fitness of such heterozygotes does not appear to

be different from their parental accessions, which has been attributed to microbial load inducing SA, and hence, reducing growth to a similar extent in the parents as in the hybrid progeny [14].

Here, we sought to further reveal the contribution of hyperactive *ACD6* alleles to plant fitness by characterizing plant phenotypes and microbial associations in both greenhouse and field conditions. Because SA is known to affect the shoot and root microbiota [21–24], we hypothesized that *A. thaliana* plants carrying the *ACD6*-Est-1 allele would have an altered commensal microbial population when exposed to natural microbial sources such as field-collected soil [25–27]. We, therefore, prepared a set of *A. thaliana* lines with hyperactive, attenuated, and absent ACD6 function, grew the plants in field-collected soil both in field and greenhouse conditions for two consecutive seasons, and monitored molecular, physiological, and microbial phenotypes at flowering. Importantly, our pioneering use of CRISPR-edited plants in the field enabled us to unequivocally ascribe phenotypes to ACD6 function in genetic backgrounds collected from the wild. We observed a dramatic display of phenotypic plasticity, with phenotypes associated with hyperactive *ACD6*-Est-1 activity in the greenhouse being suppressed in field conditions at both the organismal and molecular level.

The prevalence of *ACD6*-Est-1-type alleles in natural populations indicates that they have been maintained by selection, but under which conditions they are beneficial in nature remains elusive. This study not only highlights the risk of studying plant immune system components only in controlled conditions but also demonstrates the value of CRISPR-edited organisms in fundamental studies of evolutionary biology and ecology. We also conclude that a better understanding of the environmental regulation of growth-defense trade-offs could improve yield in crops that use *ACD6*-dependent signaling as part of their defense repertoire. More broadly, these results serve as a stark reminder that even widely used artificial laboratory conditions may be outside the norm for an organism that evolved outdoors, be it a plant, a fungus, a worm, a fly, a mouse, or anything else, and that caution is needed when extrapolating from the laboratory to the real world.

## Results

### Phenotypes of *ACD6*-Est-1 plants in the field

To reduce confounding by genetic background without the use of transgenic plants, we took advantage of a heterogeneous inbred family (HIF) derived from an Est-1 x Col-0 cross [28] that was only segregating in the genomic interval around *ACD6* [12]. For simplicity and to minimize bias, we only planted progeny from the heterozygous HIF parents, with the mendelian expectation of progeny containing approximately 25% individuals homozygous for the *ACD6*-Est-1 allele and 25% homozygous for the standard *ACD6*-Col-0 allele. In addition, we generated transgene-free CRISPR/Cas9 genome-edited Est-1 lines in which *ACD6* was disrupted by a single-base pair deletion [11], resulting in a severely truncated open reading frame that encodes only the first 37 residues of the 671-amino acid ACD6 protein. The two contrasts allowed for comparison of *ACD6*-Est-1 and *ACD6*-Col-0 alleles in a mixed, but isogenic Col-0/Est-1 background, and of plants with the *ACD6*-Est-1 allele and an *acd6* knockout mutation (from here on: Est-1:*acd6*-null) in a pure Est-1 background.

We then planted the HIF:Est-1, HIF:Col-0 and Est-1 wild-type genotypes in late fall of 2016 in the greenhouse and at a research site of the University of Tübingen in Tübingen, Southern Germany, in the same batch of topsoil sourced from the Tübingen site. In the greenhouse only, we grew in addition Est-1:*acd6*-null plants in parallel, since genome-edited plants are subject to GMO rules in the EU and, therefore, cannot be easily grown outdoors without a lengthy permitting process. Instead, we grew Est-1 and Est-1:*acd6*-null plants outdoors in 2019 at a secure facility at Agroscope in Zurich-Reckenholz, Switzerland, using field soil from that site. We started plants from stratified seeds, and the outdoor plants overwintered in an open field without further protection (Methods, S1 Fig). To harvest faster-maturing greenhouse plants at a similar phenological stage, we started them in late January the following year using remaining homogenized soil that had been left over winter in the field; the greenhouse plants, therefore, did not pass through a cold vernalization period.

PLOS Biology

When both indoor and greenhouse plants began to flower at the end of March, we took overhead photos and quantified green pixels of their rosettes (Methods, S1 Fig) as a proxy for biomass [29]. Greenhouse plants showed typical *ACD6*-Est-1 phenotypes, with HIF:Est-1 and Est-1 being noticeably smaller and more necrotic than HIF:Col-0 and Est-1:*acd6*-null plants (Fig 1A, 1B). Unexpectedly, in the field, late-onset necrosis in HIF:Est-1 and Est-1 was not apparent, and these genotypes were no smaller than their isogenic counterparts with the standard *ACD6*-Col-0 allele or an *acd6* knockout mutation. A repeat planting of the same genotypes in 2017 produced similar results. The following in-depth analyses are mostly based on plants from the first planting season.

### *ACD6* drives a strong immunity transcriptional program only in the greenhouse

The lack of visual phenotypes linked to *ACD6* genotype in field conditions suggested that *ACD6* activity is strongly dependent on growth conditions. We, therefore, extracted mRNA and sequenced transcriptomes from all genotypes in both the field and greenhouse for the 2016 German plants and the 2018 Zurich plants. To avoid transcriptional variance due to circadian factors, we harvested rosettes in a narrow time window, working quickly to minimize wounding responses (Methods). After constructing mRNA-seq libraries with a custom protocol [31], we mapped reads to the Col-0 TAIR10 reference genome (TAIR, http://arabidopsis.org).

We first examined broad-scale differences between the samples through principal component analysis (PCA) ordination, using the 500 genes with the greatest variance across genotypes and environments. Gene expression differed greatly by environment, with greenhouse and field plants cleanly separated (Fig 2A). mRNA levels also differed, as expected, by background genotype, with HIF plants, which have a mixed Col-0/Est-1 background, being distinct from plants in a pure Est-1 background (Fig 2A). In the greenhouse, but not in the field, plants with the hyperactive *ACD6*-Est-1 allele were separated in the PCA from their counterparts with the standard or knockout *ACD6* alleles (Fig 2A). In total, greenhouse conditions led to significant upregulation of 296 genes in HIF:Est-1 as compared to HIF:Col-0, and 305 genes in Est-1 as compared to Est-1:*acd6*-null (FDR-adjusted Wald test, $P<0.01$). While only 80 upregulated genes overlapped between HIF:Est-1 and Est-1 (Fig 2B), this group of upregulated genes was mostly involved in defense signaling pathways, as inferred from Gene Ontology (GO) enrichment analysis (Fig 2C). This is consistent with the previously observed induction of defense marker genes in plants with *ACD6*-Est-1 and other high-activity *ACD6* alleles [10,12,18]. In stark contrast to the greenhouse, the number of upregulated genes in field plants with the *ACD6*-Est-1 allele was only in the single digits, and they did not overlap between the two genetic backgrounds, and the handful of genes that did differ from the control plants had no known links to the immune response. In a complementary analysis independent of arbitrary significance thresholds, we also analyzed expression of the top 100 genes in each environment, either the 100 genes with the lowest *P*-values (S3A Fig) and those with the highest fold-changes (S3B Fig). In both cases, we saw a similar pattern, where there was substantial overlap only for allelic contrasts in the greenhouse conditions, with enrichment for defense pathways (S4 Fig)

Zooming in on individual genes, we first verified that the commonly used control *UBQ10* [32] was consistently expressed across our samples (Fig 2D). In both the greenhouse and the field, *ACD6* transcripts in Est-1:*acd6*-null were less abundant than in Est-1, despite the mutation being in the coding sequence, consistent with both nonsense-mediated decay as well as positive feedback regulation of *ACD6* [9,33] (Fig 2D). Expression of *ACD6* was also lower in the field compared to the greenhouse, particularly in the pure Est-1 background (Fig 2E). In the HIF background, expression of *ACD6*-Col-0 and *ACD6*-Est-1 alleles was similar for both genotypes, consistent with the results of previous analyses in which either allele had been introduced as a transgene in the Col-0 background [12]. The slightly lower apparent expression of the *ACD6*-Est-1 allele in the HIF background was not significant and could be explained by reduced read mapping of the divergent *ACD6*-Est-1 allele to the Col-0 TAIR10 reference.

Classic defense marker genes showed strong differential regulation; the SA-responsive marker gene *PR1* (Fig 2D) and the jasmonic-acid (JA) marker *PDF1.2* (Fig 2E) were greatly upregulated in the greenhouse, but not in the field, in both

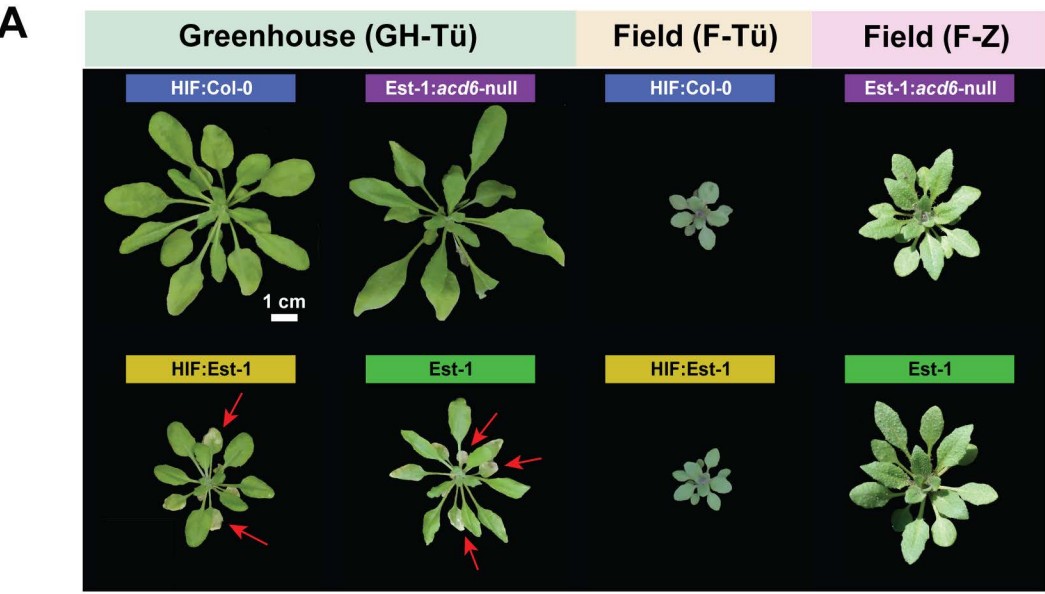

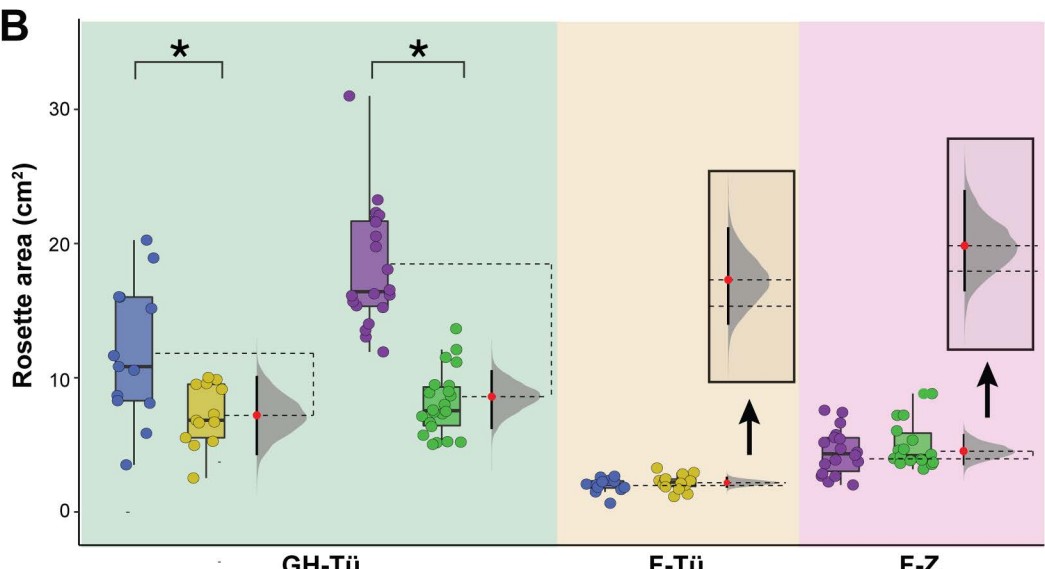

**Fig 1. A hyperactive *ACD6* allele reduces growth and causes necrosis only in the greenhouse. (A)** Representative *Arabidopsis thaliana* rosettes from different genetic backgrounds. Yellow and blue labels represent descendents of a heterozygous *ACD6* HIF line that are homozygous for either the Est-1 or the Col-0 allele of *ACD6*. Green and purple labels represent the Est-1 accession and the isogenic *acd6* knockout null allele in that accession. Plants were grown in Tübingen in the greenhouse (GH-Tü), outdoors in the field in Tübingen (F-Tü), or outdoors in the field in Zurich (F-Z, a GMO-approved facility). Matching field soil was used, plants were watered from the top to simulate the effects of rain, and no supplemental light was used in the greenhouse. All plants are shown at the same scale (scale bar 1 cm). Note the necrotic older leaves for greenhouse plants with the *ACD6*-Est-1 allele (red arrows). **(B)** Rosette size of individual plants, calculated by quantifying green pixels from images similar to those in panel A. The line within each box represents the median. Boxes enclose the interquartile range (IQR) with whiskers extending to up to 1.5 times the IQR. Brackets indicate Mann–Whitney *U*-test results, with * indicating $P < 0.05$ after Benjamini–Hochberg correction for multiple testing across the four comparisons. Dotted lines extending from the mean of each distribution span the observed effect size, and a bootstrapped distribution of computed effect sizes [30] is shown in gray, centered on the mean of the second box in each pair (red dot). The black bar extending vertically from the red dot represents the 95% confidence interval of the effect size. The data underlying this figure can be found in https://doi.org/10.5281/zenodo.15527338.

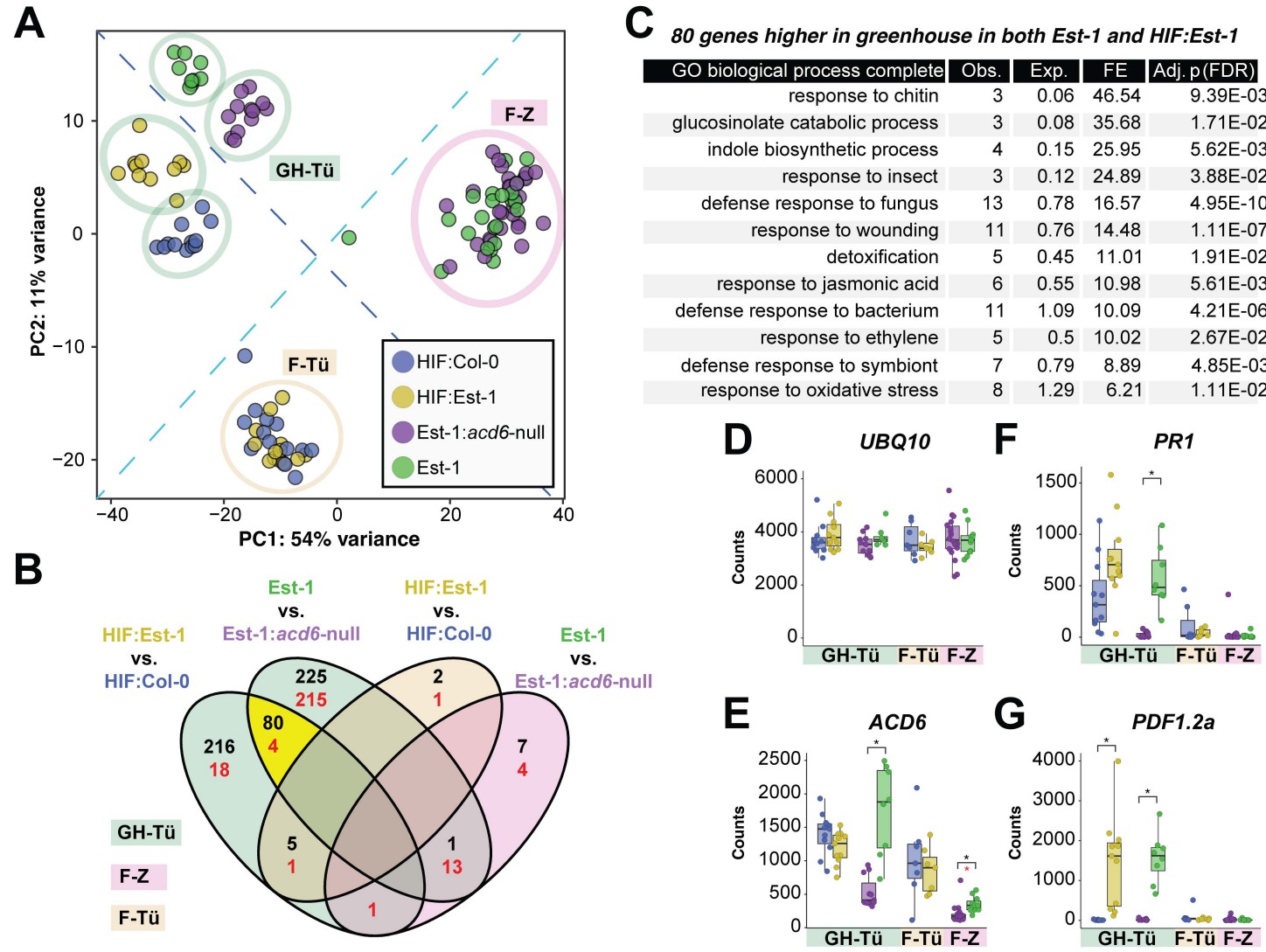

**Fig 2. A hyperactive *ACD6* allele upregulates defense responses specifically indoors. (A)** Principal component analysis of mRNA counts calculated from the 500 genes with the most variance across genotypes and environments. Samples are colored by plant genotype as indicated by the legend. Circles indicate the environments (GH-Tü = Greenhouse Tübingen, F-Tü = Field Tübingen, F-Z = Field Zurich). The dashed light blue line separates field and greenhouse environments, and the dashed dark blue line separates *ACD6* allelic contrasts. The color code applies also to other panels. **(B)** Overlaps between upregulated (black numbers) and downregulated (red numbers) genes in *ACD6* allelic contrasts, either in the greenhouse (green) or field (sand and pink), at a $P < 0.01$ significance threshold after adjusting for multiple testing. The overlap region highlighted in yellow is further analyzed in the next panel. **(C)** The gene ontology (GO) processes overrepresented in the 80 genes upregulated in the greenhouse for both *ACD6* allelic contrasts. Obs., Observed count; Exp., Expected count; FE, Fold enrichment; Adj. *P* is the FDR-corrected *P*-value. **(D–G)** Normalized mRNA counts for the *UBQ10* reference gene, *ACD6*, the SA marker *PR1*, and the jasmonic-acid (JA) marker *PDF2.1a*. Boxes enclose the interquartile range (IQR) with whiskers indicating 1.5 times the IQR. * signifies a $P < 0.01$ in an FDR-corrected Wald test and $P < 0.05$ in an independent Mann–Whitney *U*-test after Benjamini–Hochberg correction for multiple testing over the four comparisons. The data underlying this figure can be found in https://doi.org/10.5281/zenodo.15527338.

genetic backgrounds with *ACD6*-Est-1 alleles, consistent with *ACD6*-Est-1 increasing levels of both SA and, more moderately, JA [12]. In the greenhouse, several other marker genes were also more strongly expressed in *ACD6*-Est-1 plants in the greenhouse, but not in the field (S2 Fig).

We quantified several major plant hormones in greenhouse and field plants by LC–MS. Levels of both SA and its inactive vacuolar storage form SA-2-O-β-D-glucoside (SAG) were higher in greenhouse-grown than in isogenic field-grown plants, and in the greenhouse, more abundant in Est-1 wild type than in Est-1:*acd6*-null plants, similar to expression of the SA marker gene *PR1* (Figs S5 and 2). SA and SAG concentrations were not noticeably different in greenhouse-grown HIF:Est-1 plants compared to HIF:Col-0 plants, consistent with the more modest differences in *PR1* expression between these two genotypes in our experiments. JA accumulated to higher levels in greenhouse-grown than in field-grown plants, and also in greenhouse-grown Est-1 and HIF:Est-1 plants compared to Est-1:*acd6*-null and HIF:Col-0 plants, respectively, mirroring the expression of JA marker gene *PDF1.2a* (Figs S5 and 2). Indole-3-carboxylic acid (ICA), a tryptophan-derived indolic hormone implicated in defense [34], showed a similar pattern to JA with increased levels in greenhouse-grown plants carrying *ACD6*-Est-1 and relatively lower accumulation in field-grown plants. Indole-3-acetic acid (IAA), the only hormone tested with merely peripheral roles in defense, was also the only hormone with higher levels in field-grown plants, and the only that appeared unaffected by *ACD6* genotype.

## The Est-1 allele of *ACD6* does not create an obvious fitness liability in the field

The differences in transcriptomes, plant hormone levels, and both plant size and late-onset necrosis between plants with and without *ACD6*-Est-1 strongly suggested that there was a sustained higher level of *ACD6* activity in the greenhouse not only at harvest but also throughout the plant life cycle, which should affect plant fitness. The lack of clear *ACD6*-dependent phenotypes in the field plants, however, did not exclude the possibility that *ACD6* was nonetheless affecting field plant fitness, e.g., by affecting seedling survival or pollination success. This concern was especially relevant because our harvest time point at the onset of flowering was not suited to address any role *ACD6* might play during later stages of the life cycle, including flowering and seed set.

To determine the contribution of *ACD6* alleles to reproductive fitness, we collected first-generation seeds (G1) from HIF parents heterozygous for the *ACD6*-Col-0 and *ACD6*-Est-1 alleles in three starting batches. As a control, we first germinated seed aliquots of each in bulk on agar, extracted DNA from pooled seedlings and determined allele ratios by amplicon sequencing; this analysis confirmed that the germinated G1 seedlings had equal representation of *ACD6* alleles (Methods, Fig 3B). We next planted hundreds of G1 seeds at high density in plastic trays and germinated them in the greenhouse and also in two outdoor field environments around Tübingen, allowing all survivors to grow to maturity. After collecting second-generation (G2) seeds in bulk, we determined the *ACD6* allele ratio in the viable seeds by germinating G2 seeds on agar and measuring the *ACD6* allele ratios (Fig 3B). In the field, the ratio of *ACD6*-Est-1 to *ACD6*-Col-0 alleles remained balanced. However, in the greenhouse, the *ACD6*-Col-0 allele was twice as common as the *ACD6*-Est-1 allele, demonstrating a clear and strong fitness penalty for the *ACD6*-Est-1 allele in the greenhouse (Fig 3C), consistent with the considerably smaller stature in that environment (Figs 1 and 2).

## Temperature alone does not explain the difference between greenhouse and field conditions

A sustained greenhouse temperature of 23°C is higher than the average temperature *A. thaliana* usually experiences in the field during its main growth period [20,35]. Further, many ecologically relevant phenotypes in *A. thaliana*, including the activity of the immune system, are often affected by shifts between 23 and 16°C [13,36,37]. We, therefore, considered that the sustained higher temperatures in the greenhouse might induce the deleterious effects of *ACD6*-Est-1, even though lower temperature normally enhances rather than ameliorates immunity-dependent phenotypes [19]. However, in controlled conditions, at both 23 and 16°C, we observed clear developmental differences between Est-1 and Est-1:*acd6*-null plants (S6 Fig). We conclude that environmental factors other than simply a change in temperature are responsible for the phenotypic differences in expression of the *ACD6*-Est-1 phenotype between greenhouse and the field.

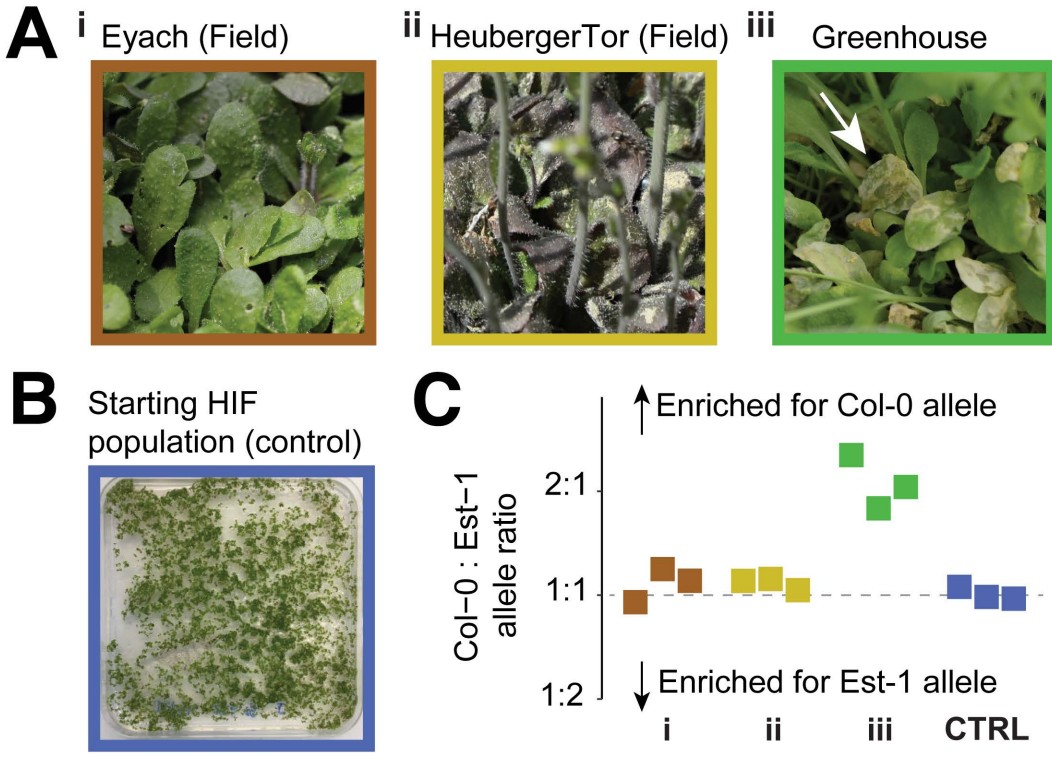

**Fig 3. The fitness penalty of *ACD6*-Est-1 is limited to the greenhouse. (A)** Leaves of HIF progeny raised to maturity at high density in different environments. (i) Flowering plants in the field at Eyach, with some leaf holes due to insect herbivory. (ii) Flowering plants in the field at Heuberger Tor, with darker leaf colors due to anthocyanin production. (iii) Flowering plants raised in the greenhouse. Note the necrotic leaves (arrow) characteristic of plants carrying *ACD6*-Est-1. **(B)** Seedlings grown from the same batch of seeds used in A, germinated on agar as a control to capture the allele frequency in the starting population. **(C)** Ratio of *ACD6*-Col-0 to *ACD6*-Est-1 alleles recovered in bulk from freshly germinated viable seeds collected from mature plants from A or from the starting population in B. Colors represent different environments as indicated by the picture border colors in panels A and B. The vertical dashed line represents a balanced allele ratio of 1:1. The data underlying this figure can be found in https://doi.org/10.5281/zenodo.15527338.

### *ACD6*-Est-1 does not affect the foliar microbiome in the field

While our molecular, metabolic and phenotypic analyses did not reveal any obvious differences between mature plants with and without the *ACD6*-Est-1 allele in the field, we hypothesized that this could have been due to *ACD6*-Est-1 only acting in early seedlings or intermittently in field conditions. If this were the case, there might be a footprint in the microbiota that colonizes plants with an *ACD6*-Est-1 allele.

For the analysis of the bacterial microbiota, we prepared and sequenced bacterial amplicons of the V4 region of the 16S rRNA region (hereafter rDNA) from total leaf DNA extracted from rosettes of field- and greenhouse-grown rosettes n Tübingen in 2016–2017, a biological replicate in Tübingen 2017–2018, and a single field replicate of the Est-1 versus gene-edited Est-1:*acd6*-null lines grown outdoors in Zurich in 2018–2019. We compared the results with previously generated V4 16S rDNA data from the phyllospheres of wild *A. thaliana* populations collected near Eyach in the Tübingen region during a previous season [38]. For fungi, we generated internal transcribed spacer (ITS) data from rosettes collected in Tübingen in 2016–2017, comparing these previously generated ITS data from the natural Eyach population [38].

We first classified amplicon sequence variants (ASVs) from all datasets to the family level and clustered samples by similarity. Unexpectedly, the bacterial microbiomes of the plants from the German field experiment were much more similar to that of wild *A. thaliana* plants with different genetic backgrounds, growing in diverse soils, being of a wider

range of rosette size and developmental status and harvested in a different year than they were to the microbiomes of isogenic greenhouse plants grown in the same soil and harvested at the same time (Fig 4A). This was also the case for fungal microbiomes (Fig 4B), demonstrating the smaller role of soil compared to other environmental factors in structuring outdoor foliar microbiomes. When we looked specifically in field plants for ASVs that differed in abundance between HIF:Est-1 and HIF:Col-0 plants, or in Est-1 and Est-1:*acd6*-null plants, we found none. Even in the greenhouse, using the very same plant material for which we had already demonstrated substantial differences in expression of immunity markers due to *ACD6* alleles (Fig 2), we failed to find differentially abundant bacterial or fungal ASVs that distinguished plants with or without *ACD6*-Est-1. This was particularly surprising because SA, the levels of which are highly sensitive to *ACD6* activity [13,14,17], has been shown to alter the microbiota in many other settings—although these effects are clearer in roots than in leaves [21–24].

We next reasoned that, perhaps, the broad-scale immune response triggered by *ACD6* might reduce the colonization of all microbes rather than select taxa, and we, therefore, compared the ratio of microbial sequencing reads to plant reads in whole-genome shotgun data [38] or hamPCR data [39] to calculate microbial load in our samples (Methods). While this revealed field plants to have approximately double the bacterial load as greenhouse plants, there were no significant *ACD6*-dependent differences in microbial load either in the greenhouse or the field (Fig 4C).

Given these negative results, we reasoned that *ACD6*-dependent defenses may be able to restrict specifically the growth of invasive opportunistic microbes without affecting the commensal microbes that colonize undamaged surfaces. We, therefore, conducted *Pseudomonas* infections in both the field and a growth chamber. For the field infection, we grew overwintered HIF plants outdoors in Tübingen, sprayed them in the spring prior to bolting with a mixture of four closely related *Pseudomonas viridiflava* strains that share the same 16S rRNA gene sequence and that occur naturally on *A. thaliana* around Tübingen ("Pv-ATUE5" strains) [40,41], and harvested plants 1 week later. As in all other experiments with field-grown plants, we noticed no *ACD6*-dependent differences in size or in late-onset necrosis. This time, unlike prior harvests, we then surface-sterilized all rosettes as in [42] to remove any transient or loosely unassociated microbes, and sequenced 16S rDNA amplicons using hamPCR to also determine bacterial load [39]. Again, we found no differences in bacterial associations with HIF:Est-1 versus HIF:Col-0 homozygotes, including for the ASV corresponding to Pv-ATUE5 (S7A Fig), and we further observed no *ACD6*-dependent difference in bacterial load (S7B Fig). For the growth chamber infection, we grew Est-1 and Est-1:*acd6*-null plants for 5 weeks in short days in a growth chamber and used a blunt-ended syringe to directly inoculate leaf apoplasts with either *Pseudomonas syringae* pv. tomato DC3000 (Pst DC3000), an uncharacterized mixture of cultured *A. thaliana* phyllosphere microbes ("other" microbes), both Pst DC3000 and "other" microbes, or vehicle (MgCl$_2$). We harvested 3 days post-infection and determined bacterial load and composition with hamPCR. Here, while there was no difference in total microbial abundance caused by *ACD6* genotype (S7C Fig), we did observe genotype-dependent differences in abundance for a handful of the more abundant ASVs (Fig 4D). Importantly, this included the ASV corresponding to Pst DC3000, which had a significantly higher load in Est-1:*acd6*-null than in Est-1 plants (Figs 4D and S7E), consistent with reduced immune function when ACD6 function is lost completely [9,11]. For plants sprayed only with Pst DC3000, we also enumerated Pst DC3000 colony-forming units; this orthogonal approach also revealed, in support of the hamPCR data, a slightly higher pathogen load in Est-1:*acd6*-null plants (S7D Fig). Besides Pst DC3000, other ASVs, including ones corresponding to *Ralstonia* sp. and *Robbsia* sp. were also significantly more abundant in Est-1:*acd6*-null plants.

## Discussion

The *ACD6* gene was originally characterized through a lab-induced gain-of-function *acd6*-1 mutation that increases levels of SA, spontaneous cell death, and resistance to pathogens but also reduces plant stature [17]. *acd6*-1 is an example of lesion mimic mutants, which exhibit disease-like symptoms in the absence of pathogens [43]. Induced lesion mimics not only provide insight into gene function, plant immunity, and plant physiology, but they are also useful to understand

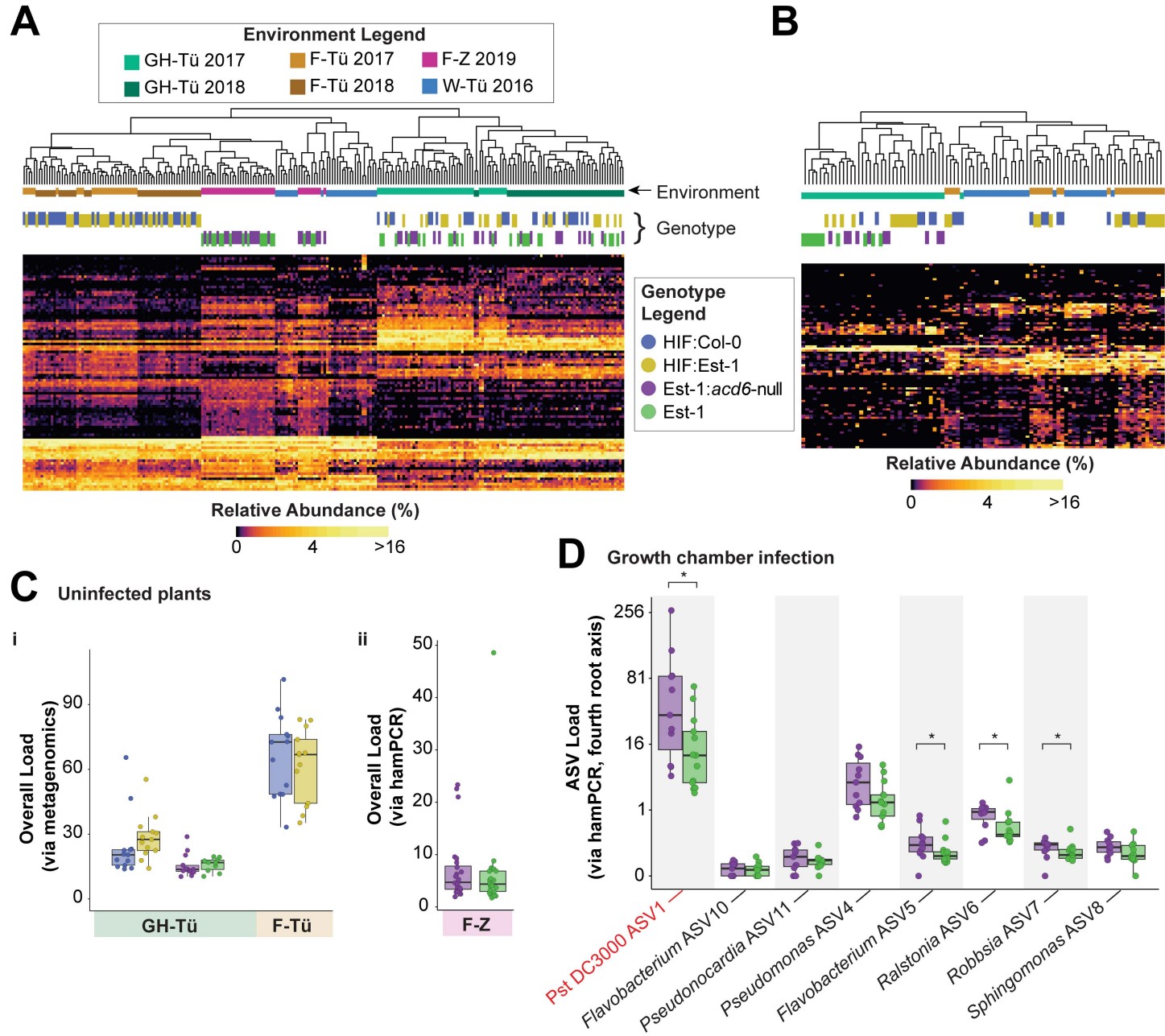

**Fig 4. *ACD6*-Est-1 has negligible effects on colonization by a natural microbiome. (A)** Heatmap showing square root transformed relative abundances of bacterial families (rows, unlabeled) based on classification of V4 16S rDNA bacterial reads from both planted *A. thaliana* individuals and plant samples from wild populations (columns in heatmap). Samples in the heatmap are clustered by Bray–Curtis dissimilarity, with groupings shown in the dendrogram. The environment of cultivation is indicated by a horizontal color key (vertically staggered only to improve visibility) between the dendrogram and heatmap, using colors in the "environment legend." The genotype of the plants is similarly shown using colors in the "genotype legend." For the environment legend, Gh-Tü = greenhouse Tübingen; F-Tü = field Tübingen; F-Z = field Zurich; W-Tü = wild Tübingen (natural populations near Tübingen). **(B)** Same as A, but showing fungal families based on ITS2 amplicons. **(C)** Bacterial load, i: for Tübingen-grown plants from 2017 as calculated by the ratio of bacterial reads to plant reads in metagenome data. ii: for Zurich-grown plants from 2019 as deduced from hamPCR. Environment and genotype colors follow the same code as in A. Boxes enclose the interquartile range (IQR) with whiskers extending to up to 1.5 times the IQR. **(D)** Bacterial loads of the most abundant ASVs in growth chamber-grown plants infected with Pst DC3000 and a mix of uncharacterized phyllosphere microbes. *$P < 0.05$ in a Mann–Whitney $U$-test with Benjamini–Hochberg correction for multiple testing of each of the eight ASVs. The ASV corresponding to Pst DC3000 is labeled in red. The data underlying this figure can be found in https://doi.org/10.5281/zenodo.15527338.

trade-offs between growth and disease resistance. An important question then is whether natural lesion mimics might confer long-term evolutionary benefits. Results with induced lesion mimics have provided mixed clues. For example, *cpr5*, which has constitutively increased SA levels, has also greatly reduced fitness in field conditions [44]. What is fascinating about the *ACD6* locus is that natural alleles that induce a hyperactive immune system, such as Est-1 type alleles, are maintained in wild populations at substantial frequencies, about 10%–20% [12,13,15]. This is indicative of *ACD6* allelic variation being under balancing selection in nature—a hallmark of many though not all immune genes [45–50]. Given the documented roles of *ACD6* in immunity under greenhouse conditions, the most parsimonious explanation is that balancing selection at *ACD6* is best explained by increased defense outweighing the drawbacks of reduced growth. We envisaged a simple scenario, wherein a natural hyperactive *ACD6* allele would result in increased SA levels in field plants, just as it does in greenhouse plants, and that this would lead to obvious phenotypic consequences in the field. However, the reality appears more complex, as we have not identified any field phenotype for selection to act upon.

In our field experiments, we could neither observe a benefit nor a cost of the *ACD6*-Est-1 allele, and we observed no major differences between HIF lines that differed in *ACD6* genotype in either life-time fitness or markers of immune system activation. Although we were not able to measure field fitness in Est-1:*acd6*-null plants, which would require letting plants set seeds, the fact that we did not observe any other phenotypic difference between Est-1 and Est-1:*acd6*-null plants in the field in Zurich suggests that life-time fitness was also unaffected even when compared to a complete loss of *ACD6* activity. This sets our study apart from work with *A. thaliana* lines that carry deletions for individual disease resistance genes encoding nucleotide-binding site leucine-rich repeat receptors, where fitness differences outdoors are substantial, despite the absence of obvious morphological differences in either the greenhouse or the field [16,51].

In support of our conclusions, it was previously shown that heterozygous combinations of certain *ACD6* alleles in the same plant, including combinations that naturally occur in hybrid zones, can lead to severe hybrid necrosis in the greenhouse, but that these are partially attenuated in field conditions [13,14]. Therefore, we speculate that *ACD6* hyperactivity in nature is very context dependent, being expressed either only during very specific developmental stages or only under very specific environmental conditions.

We do not know yet what aspect of the greenhouse environment results in such strongly variable expression of *ACD6* hyperactivity. One obvious possibility is that the controlled greenhouse has less abiotic stress such as drought as well as fluctuating insolation and temperature; abiotic stress is known to suppress plant immune responses [52]. Indeed, that the greenhouse had more favorable conditions can be inferred from more plants growing more quickly and much larger (Fig 1). The most upregulated mRNA GO terms in the field compared to the greenhouse were "response to light intensity" and "carotenoid synthesis," so it is plausible that light stress in the field was a factor in suppressing SA pathways [53]. Another possibility is that greater outdoor UV-B radiation prompted greater production of protective secondary metabolites, reducing the need for SA-mediated pathways [53]. A further possibility is that *ACD6* hyperactivity is growth rate dependent, with the slower growth outdoors greatly dampening *ACD6* hyperactivity. In this scenario, *ACD6* only comes into play under circumstances where plants encounter very favorable circumstances for growth. Yet another possibility is that one or more members of natural microbial populations on the field plants suppress SA-mediated defenses, including *ACD6* [54].

It is important to note that in natural accessions, the activity of *ACD6* alleles depends on variation at other loci [12,15]. The small proteins MHA1 and MHA1-like (MHA1L) can interact with ACD6 to suppress or enhance its activity [11]. The full details of how such cryptic modifiers of *ACD6* activity respond to fluctuating environmental cues are unknown, and these loci could be important in the phenotypic plasticity observed here.

We also tested the hypothesis that transient hyperactivity of *ACD6*-Est-1 in the field might have lasting effects on the foliar microbiome and, thus, might provide indirect benefits in the event of pathogen attack [55,56]. Using amplicon sequencing, we did not observe differential natural microbial colonization across *ACD6* genotypes, even in greenhouse conditions where *ACD6*-dependent phenotypes were very clear and conditions for which an A*CD6*-dependent reduction

in accumulation of pathogenic microbes has been previously documented [12]. Using both shotgun sequencing and hamPCR, we also did not observe an effect of *ACD6* on bacterial load, even in the greenhouse. We addressed this confusing situation by inoculating greenhouse plants with commensal microbes in the presence of a bacterial pathogen and looking at both pathogen and commensal abundance via amplicon sequencing. Both colony counting and hamPCR revealed a slight but statistically significant reduction in the proliferation of Pst DC3000 in Est-1 compared to Est-1:*acd6*-null plants, consistent with previous observations [9,11,57]. However, few other ASVs seemed to be affected. Our data strongly suggest that, despite very high SA levels, effects of the *ACD6*-Est-1 allele on commensal phyllosphere bacterial and fungal microbes are slight. To date, few studies have addressed the role of SA levels on foliar microbes, and the clearest effects have been observed in hyperimmune mutants with extreme phenotypes grown in gnotobiotic conditions with a synthetic community of bacteria [23,58]. On the other hand, knocking out receptors for microbe-associated molecular patterns does not necessarily have major effects on the endophytic microbiome in field grown plants either [59].

The negative impact of the natural *ACD6*-Est-1 allele on the seed number of greenhouse-grown plants was severe. Should such *ACD6* alleles be activated at the wrong times in field conditions, even slightly, this could have significant effects on fecundity. Knowing the triggering environmental conditions, how they modulate *ACD6*-Est-1, and how important different *ACD6* alleles are for fitness during epidemics could have practical relevance for major cruciferous oilseed crops like *Brassica napus*, for which *ACD6*-orthologs can be identified bioinformatically, and for which there is a wide bank of potential breeding germplasm. We propose that the suite of broad-spectrum defenses that can be upregulated by *ACD6* under certain conditions, which are apparently not common outdoors but the norm in the greenhouse, behave like a "speed bump" in traffic, which poses little hindrance to vehicles maintaining the speed limit, but which seriously hinders or damages vehicles traveling too fast. Similarly, commensal microbes that grow slowly without damaging the plant may be able to tolerate increased intracellular concentrations of antimicrobial peptides and secondary metabolites that they rarely encounter, but pathogens that rely on extraction of plant resources for fast growth might be much more sensitive to such defenses.

Future work will be necessary to determine what it is in the controlled conditions that brings out these cryptic and otherwise deleterious *ACD6* phenotypes. If it is some combination of abiotic factors that could be easily replicated in the laboratory, this may precipitate a paradigm shift in standard growth conditions for *A. thaliana*, which, in turn, could improve generalizability of fundamental laboratory plant research. In either case, the results of our study provide an important lesson: even when robust phenotypes are reproducible across different labs, they are not necessarily indicative of what happens in the real world.

## Methods

### Plant material and seed treatment before planting

*A. thaliana* accessions and HIF lines were derived from stocks maintained in the lab. All seeds were surface-sterilized by a 1 min submersion in 70% ethanol with 0.01% Triton X-100, followed by a 12 min submersion in over-the-counter bleach (DanKlorix; 2.8% NaOCl w/w) diluted to 10% in water, followed by three washes in sterile distilled water. Seeds were left in water and stratified at 4°C in sterile centrifuge tubes for 1 week before planting.

For genome editing, an *A. thaliana* codon-optimized Cas9 (*athCas9*) [60] was used, with the final *pUBQ10::athCas9:trbcs::gRNA::mCherry* constructs assembled from six GreenGate modules [61]. The same gRNA was encoded on the transgene by 5′-GTGTC GCCCG TAGGT GACG-3′ for both ACD6-Col-0 and ACD6-Est-1. The primers for generating the gRNA cassette were 5′-GTGTC GCCCG TAGGT GACGG TTTTA GAGCT ATGCT GAAA-3′ and 5′-CGTCA CCTAC GGGCG ACACC AATCA CTACT TCGAC TCTA-3′. Red fluorescence from a *pAT2S3::mCherry:tMAS* cassette [62] was used for selection of transgene-free seeds. The edited lines were whole-genome-sequenced using 2 × 150 bp reads on the HiSeq3000 platform (Illumina).

## Cultivation in natural soil for phenotyping, RNA-seq, phytohormone measurements, and microbiome profiling

**Field in Tübingen, Germany.** Topsoil was obtained from a research field of the University of Tübingen at the Heuberger-Tor-Weg in Tübingen, Germany (48°32′44.9″N 9°02′32.8″E) in October 2016 for most plants, and again in 2017 for a replicate experiment. The soil was initially spread out in plastic trays in a plastic foil tunnel and allowed to dry for 1 week until the clayey soil began to crumble to the touch, which made it easier to break larger aggregates in the soil with physical force and to sieve the material to make it more homogeneous. All soil was sieved through a 8 mm mesh to remove rocks and twigs, mixed thoroughly by repeatedly turning the pile over with a shovel (S1 Fig), and filled evenly into 40-pot quickpot "QP 40 T/11.5" plastic trays (HerkuPlast Kubern GmbH, Ering, Germany) that were placed inside flat trays to catch water from the top, either rain outdoors or from top watering. After pots were filled, the remaining soil was transferred into 10 L plastic pots for storage and overwintering outside until it could be homogenized again for use in the later greenhouse experiment (see below: "Greenhouse in Tübingen, Germany").

We pre-watered each 40-pot flat with distilled water by bottom watering and sowed surface-sterilized, cold-stratified seeds at the end of October 2016 (and again in October 2017) when daily highs were around 14°C and lows around 1°C. Stratification for 1 week prior to sowing ensured uniform germination of the cohort, coinciding with the germination of wild winter-annual *A. thaliana* plants in local stands around Tübingen. Each plant of the HIF genetic background was planted in its own pot. Since individual seeds had not been genotyped, plants were automatically randomized for *ACD6* genotype. After germination outdoors for 1 week under transparent plastic lids, the flats spent another month without lids in an open foil tunnel [63] allowing exposure to wind and ambient temperatures but protection from precipitation. Wild plants of other species that germinated, as well as excess *A. thaliana* seedlings, were removed regularly with tweezers so that a single seedling developed in the middle of the pot. Finally, after the first true leaves appeared, plants were moved to an open location where they had full environmental exposure, and the drip trays were removed to prevent water accumulation and flooding during rainfall.

**Field in Zürich, Switzerland.** For CRISPR/Cas9 mutant plants, we followed a similar protocol as above to stratify and germinate Est-1 and Est-1:*acd6*-null seeds at the end of October 2018 in local Swiss soil at Agroscope in Zurich-Reckenholz, Switzerland (47°25′40.8″N 8°31′01.1″E), in a secure open-air room. This room had a net with mesh size >1 cm$^2$ instead of a roof and an open wall of chain-link fence, and allowed for plant exposure to natural wind, rain, and insects. Instead of placing pots in flats, they were arranged on open soil growth boxes, and extra soil was filled around the pots to hold them in place. Overwintered seedlings were cleaned of weeds and thinned to 1 per pot on 21 February 2019, and finally, mature plants were harvested on 21 March 2019.

**Greenhouse in Tübingen, Germany.** Field soil used for greenhouse cultivation was prepared as described above beginning in autumn, with the portion used for field cultivation distributed into pots immediately and the portion to be used for greenhouse cultivation placed in 10 L open plastic pots for storage and overwintering. In January 2017, these 10 L pots were emptied into a single pile which was re-homogenized by turning it over repeatedly with a shovel, and the soil was then filled into 40-pot quickpot plastic trays that were, in turn, placed inside flat trays to catch water, exactly as for field cultivation. On 1 February 2017, these trays were placed in a greenhouse room without any supplemental lighting averaging 24°C daily highs and 20°C nightly lows, and seeds were planted as above. After 1 week of germination under plastic lids, the lids were removed and the greenhouse plants were top watered with distilled water from a watering can to mimic rainfall including water splash from the soil onto the plants. Mature plants were harvested on 22 March, 2017.

## Imaging and harvesting plants grown in natural soil

Both greenhouse and field plants in all locations began to develop a floral meristem at the end of March, and thus were phenologically similar at the time of harvest. We first took overhead photos of all plants using a LUMIX DMC-TZ71 digital camera (Panasonic, Osaka, Japan) without flash. Because of slight differences in the focal length between photos, all images were rescaled in Adobe Photoshop against an internal standard such that each pixel

represented the same true area. For smaller field plants where the rosettes were fully contained within the circumference of their pot, a predefined mask was used to extract the area surrounding each plant in the image, and automatic segmentation based on pixel color was applied to distinguish plant leaves from background. For larger greenhouse plants, where leaves had projected beyond the perimeter of the pot, we extracted the area surrounding each plant and manually erased leaf tips from neighboring plants in Adobe Photoshop before using automatic segmentation to recognize green pixels of the central plant in each image. Rosette area was then approximated as green pixels in each image.

In all locations, plants were harvested in a randomized fashion between 10:30 AM and 12:30 PM to minimize circadian effects. In Tübingen, field plants were harvested first, and greenhouse plants on the following day. All rosettes were first removed from their roots with scissors and tweezers that had been dipped in 95% ethanol and flame-sterilized, and rosettes were placed into 50 mL tubes for washing. Soil and dried mud (potentially containing irrelevant unassociated soil microbes) were washed off leaf surfaces by adding approximately 25 mL of non-sterile distilled water, shaking by hand, and then decanting. This was repeated until the decanted water was visually clear. Next, the rosette was washed once with sterile (autoclaved) distilled water using the same procedure. Finally, the rosette was removed from the tube with re-sterilized tweezers and placed on autoclaved paper towel for blot drying. The clean and dry rosette was loaded into a 2 mL or 15 mL screw-cap tube (Sarstedt, Nümbrecht, Germany) depending on the plant size, and snap-frozen in liquid nitrogen. The entire harvest process from cutting to freezing took place in under 3 min for each rosette to minimize transcriptional noise from harvest-related stress. Sample numbers of harvested plants used for various assays are given in S8 Fig.

## Cultivation for plant phenotyping at 23°C or 16°C

**Growth chamber in Tübingen, Germany.** After stratification, Est-1 and Est-1:*acd6*-null seeds were germinated and cultivated in potting soil in individual pots in growth rooms at a constant temperature of 23°C or 16°C, air humidity at 65%, 16-h (long days) day length. Philips GreenPower TLED modules (Philips Lighting GmbH, Hamburg, Germany) provided 110–140 µmol m$^{-2}$ s$^{-1}$ light with a mixture of 2:1 DR/W LB (deep red/white mixture with ca. 15% blue) and W HB (white with ca. 25% blue) modules, respectively. Plants were imaged daily at 14:00 h using the Raspberry Pi Automated Phenotyping Array () [29].

## Planned pathogen and microbiota inoculations

**Field in Tübingen, Germany.** Segregating HIF seeds were surface-sterilized and sown on potting soil in 40-pot quickpots in late October 2020 and grown over winter as for field cultivation in natural soil (above). In late March 2021, we sprayed the rosettes with an equal mixture of four *P. viridiflava* Pv-ATUE5 strains (Pv-ATUE5:p11c5, Pv-ATUE5:p13g4, Pv-ATUE5:p8h9, and Pv-ATUE5:p11a6) [40,41] at a combined OD$_{600}$ of 1.0. Following 1 week of growth, we harvested whole rosettes and surface sterilized the leaves following a published protocol [42] Next, the same leaves were surface sterilized with a 45 s wash with 75% ethanol [64] to lyse bacteria, followed by a 15 s wash with 2% bleach (diluted Dan Klorix brand) to destroy residual DNA, followed by three washes with sterile water. DNA was extracted from all plants, and used to genotype each HIF plant at the *ACD6* locus (described below). Heterozygotes were not analyzed further, and bacterial composition and load was quantified in the Col-0 and Est-1 allele homozygotes using hamPCR [39]. Sample numbers are given in S8 Fig.

**Growth chamber in Uppsala, Sweden.** Plants were cultivated under similar light intensities as described above for Germany, but using 8 h (short days) day length and only at a single temperature, 23°C. Surface-sterilized Est-1 and Est-1:*acd6*-null seeds were germinated on potting soil in 7 cm plastic pots, with one plant per pot. After 5 weeks of growth, four leaves per plant were inoculated by abaxial injection directly into the apoplast with a blunt-ended 1 mL syringe [12,64]. The

inoculum was either 10 mM $MgCl_2$ (vehicle), *P. syringae* DC3000 (Pst DC3000) at $1 \times 10^5$ CFU/mL [64], uncharacterized microbes from homogenized *A. thaliana* leaves at $1 \times 10^5$ CFU/mL, or a mixture of Pst DC3000 and uncharacterized microbes at a combined $OD_{600}$ of $2 \times 10^5$ CFU/mL. The uncharacterized microbes were prepared by plating a leaf homogenate on 1/10 strength LB media solid plates with 2% agarose and 100 µg/mL cycloheximide to suppress fungal growth, scraping colonies off the plates after 5 days with an inoculation loop, and resuspending the pooled colonies in 10 mM $MgCl_2$. The Pst DC3000 inoculum was prepared by resuspending bacteria grown overnight on an LB agar plate containing 100 µg/mL rifampicin. The concentration of bacterial inoculum was adjusted by measuring $OD_{600}$ and diluting appropriately, using the approximation of $OD_{600} = 0.2$ corresponding to $1 \times 10^8$ CFU/mL. Following inoculation, transparent humidity domes were kept on the plants for 24 h. For harvest, a single hole punch with diameter 5.4 mm was taken from each of the four inoculated leaves (total.93 $cm^2$) and added to a 2-mL screw cap tube containing 300 µL of 10 mM $MgCl_2$ and two 5-mm glass balls. Tissue was homogenized by grinding for 20 s at 4.0 m/s in a FastPrep 24 5G homogeniser (MP Biomedicals, Eschwege, Germany) and log-serial dilutions were plated on LB solid media with rifampicin (100 µg/mL) to enumerate CFU/mL. Sample numbers are given in S8 Fig.

### Homogenization of plant material for DNA, RNA, and phytohormone extraction

For rosettes in 15 mL tubes, we first added 5 autoclaved 5 mm hardened steel balls (VWR, Radnor, USA), re-froze the tubes with the rosettes, and shook the deep-frozen tubes by hand to break the rosettes down into particles of about 1 mm. We then held the steel balls to the cap using an external magnet, and tapped approximately 0.25 mL of the frozen plant powder into an open and pre-frozen 2 mL screw-cap tube. To these 2 mL tubes, and also to rosettes already in 2 mL tubes, we added 0.5 mL of 1 mm garnet rocks (BioSpec Products, Bartlesville, USA) and a 5 mm glass ball (Carl Roth, Karlsruhe, Germany) and immediately re-froze them to prepare for bead beating. Frozen plant tissue was then ground "dry" in a FastPrep 24 5G homogeniser (MP Biomedicals, Eschwege, Germany) at 4 m/s for 20 s, and we immediately re-froze the resulting frozen powder.

### DNA and RNA co-extraction

To approximately 250 mg of frozen plant powder, prepared as described above, we added 750 µL of RNA/DNA lysis buffer (100 mM Tris pH 8.0, 100 mM NaCl, 10 mM EDTA, 1.5% SDS, 2% 2-mercaptoethanol, and 100 µg/mL Proteinase K) that had been pre-warmed to 50°C to enable the buffer to reach all frozen plant particles and to inactivate RNAses. We then did a final high-speed "wet" bead beating at 6 m/s for 60 s to lyse all cells. The tubes were centrifuged at 10,000$g$ for 5 min. Of the approximately 600 µL supernatant, 450 µL was transferred to a 1.5 mL centrifuge tube for DNA prep, while 150 µL was transferred to another 1.5 mL centrifuge tube for RNA extraction. For DNA extraction, the 450 µL lysate was mixed with 150 µL sterile 5 M potassium acetate in a 1.5 mL centrifuge tubes to precipitate the SDS. The tubes were spun at 10,000 $g$ for 5 min and the supernatant transferred to a new 1.5 mL tube. The resulting supernatant was centrifuged a second time to clear out remaining plant material and precipitate, and 600 µL was transferred to a 1 mL deepwell plate (Mettler Toledo, Gießen, Germany). Finally, 360 µL SPRI beads were added to 600 µL of the supernatant (0.6: 1 ratio). After mixing and incubating on a 96-well Magnet Type A (Qiagen, Hilden, Germany), the beads were cleaned with 80% ethanol and DNA was eluted in 100 µL EB (10 mM TRIS, pH 8.0).

### Phytohormone measurements

Between 50 and 150 mg of frozen plant powder was shipped on dry ice to the Mass Spectrometry Core laboratory at the University of North Carolina. LC-MS quantification was performed using a PE Sciex 3000 mass spectrometer equipped with a CTC autosampler and Shimadzu LC system. Sample numbers are given in S8 Fig.

 

## 16S rRNA and ITS amplicon sequencing

Amplicons for microbial profiling were prepared and processed as described in [38]. Plants grown in Zurich natural soil, plants grown for outdoor inoculation with Pv-ATUE5 in Tübingen, and plants grown in Uppsala for pathogen inoculation were profiled using hamPCR [39]. Sample numbers are given in S8 Fig.

## RNA-seq

Between 500 and 1,000 ng total RNA was used to construct libraries as described in [31]. Briefly, we used oligo dT) beads to purify plant mRNA, performed first- and second-strand cDNA synthesis, added adapters by ligation, and amplified the library molecules with 12 cycles of PCR before sequencing with 150 bp short reads on an Illumina HiSeq 3000 instrument. The TAIR10 reference genome was prepared for mapping using the *rsem-prepare-reference* function of RSEM [65], and reads were mapped with bowtie2 [66] as implemented in *rsem-calculate-expression*. RSEM mappings were converted into a count table with *tximport* [67], and the count table was analyzed with a negative binomial model using the DeSeq2 package in R [68]. All samples with at least 1 M mapped reads were analyzed, as well genes with at least 10 counts over all samples. Sample numbers are given in S8 Fig.

## Statistical analysis

For the RNA-seq data, "site" and "genotype" were the metadata variables of biological interest. Because we were interested in comparing *ACD6* allelic contrasts in each environment, and were not interested in overall genotype or site effects across the whole dataset, "site" and "genotype" factors were concatenated into a new factor that represented all combinations of site and genotype (as recommended in the documentation on interaction terms for DeSeq2 [68]). We then calculated differential expression using a one-factor model (~"sitegenotype"). DEGs for each genotype pair in each environment were identified from the DESeq2 output as those genes with a Benjamini–Hochberg adjusted $P$-value < 0.01 in a Wald test.

GO enrichments were calculated by loading gene IDs into the Panther Classification system (https://pantherdb.org/) with *A. thaliana* as reference organism, "GO biological process complete" as the annotation set, and default parameters (Fisher's exact test with FDR correction for multiple testing). The most specific biological process in each significant hierarchy are shown (e.g., "defense response to fungus" instead of "response to stress").

RNA expression for individual genes of interest was further visualized as boxplots (e.g., Fig 2D–2G). The hypothesis that plants carrying the *ACD6*-Est-1 allele had a different expression than *ACD6*-Col-0 or Est:*acd6*-null was tested for each *ACD6* allelic contrast using a two-tailed Mann–Whitney $U$-test, with Benjamini–Hochberg correction for multiple testing across the four site-by-genotype combinations.

For rosette size data, the hypothesis that plants carrying the *ACD6*-Est-1 allele were smaller was tested for each *ACD6* allelic contrast using a one-tailed Mann–Whitney $U$-test, with Benjamini–Hochberg correction for multiple testing across the four site-by-genotype combinations.

For microbiome comparisons at the ASV level, only samples with at least 1,000 microbial reads were included, and the data were normalized by converting to relative abundance through total sum scaling of microbial reads. Only ASVs with a mean abundance of at least 20 reads across all samples prior to total sum scaling were considered quantifiable (approximately 5% of ASVs). ASV abundance was compared for allelic contrasts in each environment using two-tailed Mann–Whitney $U$-tests with Benjamini–Hochberg correction for multiple testing.

## Genotyping of *ACD6*-Est-1 and *ACD6*-Col-0 alleles in HIF lines

**High throughput screening of allele ratios in pooled HIF seeds for fitness tests.** Seeds were surface sterilized for 1 min with 70% EtOH and 0.01% Triton X-100, followed by 12 min in 10% bleach, and finally three washes with sterile water. They were stratified for 1 week at 4°C, and germinated on 1% agar containing half-strength MS medium and 5 mM

MES buffer. Seedlings were germinated under 16 h light at 23°C and 65% relative humidity under cool white fluorescent light of 125–175 µmol m$^{-2}$ s$^{-1}$ and grown for 1 week, allowing cotyledons to emerge. Liquid nitrogen was then poured on the plates to snap freeze all plant material, and frozen, brittle cotyledons were raked off by spatula and pooled. The pooled plant material was further homogenized with a mortar and pestle, and an aliquot of approximately 250 mg macerate was used for DNA extraction as described for DNA in "DNA and RNA co-extraction".

The following primers were used to amplify a region of ACD6 containing polymorphisms distinguishing *ACD6*-Est-1 from *ACD6*-Col-0, and sequenced using an Illumina MiSeq 50 bp kit. 5′-<u>ACACT CTTTC CCTAC ACGAC GCTCT TCCGA TCT</u> tgtac tctta tttgg gcgca gtt-3′ (>Genotype_ACD6_F) and 5′-<u>GTGAC TGGAG TTCAG ACGTG TGCTC TTCCG ATCT</u> aactc agata tgtct atagt cagca ta-3′ (>Genotype_ACD6_R). In these primers, the lowercase region is complementary to the ACD6 template, while the uppercase region adds overhangs to which Illumina Nextera primers can bind in a subsequent PCR round. First-round PCR with the ACD6-complementary primers used the following program: 95°C for 2 min, 30 cycles of 95°C for 15 s, 60°C for 20 s, and 72°C for 30 s, followed by a final 5 min at 72°C. Illumina adapters were added as described in [38] for "Batch 3 plants". The full amplicon, including Illumina adapters, was 366 bp long and the sequence-able region was 299 bp. Fifty bp of sequencing from the forward primer was sufficient to confidently genotype the *ACD6*-Col-0 versus *ACD6*-Est-1 alleles as shown below, where lowercase letters represent the bases corresponding to the primers above and capital letters represent the amplified region:

ACD6_Col-0: tgtac tctta tttgg gcgca gtt **G**GGTG ATC**T**A **GCA**CT CATCC **T**CAAA TC
ACD6_Est-1: tgtac tctta tttgg gcgca gtt **A**GGTG ATC**C**A **AAC**CT CATCC **G**CAAA TC

**Determining genotypes of individual HIF plants at *ACD6*:** Because determining the *ACD6* genotype of individual HIF plants required destructive sampling, it could only occur after plants were grown and DNA was prepared. Because we were interested in *ACD6*-Est-1 and *ACD6*-Col-0 homozygotes but not the Est-1/Col-0 heterozygotes, this required growing and preparing high quality DNA from approximately twice as many plants as we ultimately analyzed. First, DNA was prepared from all plants with bead beating as described above. Two reverse primers, 5′-CAAAA CAAGT TTTGA TCTTA CG-3′ (acdHIF_estR G-47730) and 5′-GCCGC TTCTC AGAGC TAG-3′ (acdHIF_col R G-47731), diagnostic for Col-0 and Est-1 sequences were used individually in combination with the same forward primer, 5′-TCACT GCAAT TGCCC ATGT-3′ (ACD6_HIF_Forward G-47729), targeting a conserved region of ACD6. PCR was performed using the program: 95°C for 2 min, 30 cycles of 95°C for 15 s, 55°C for 20 s, and 72°C for 45 s, followed by a final 5 min at 72°C, and the products were separated on a 1% agarose gel. The amplification of only one primer pair was diagnostic of homozygous Est-1 or Col-0 sequences, while the amplification of both primer pairs indicated heterozygotes.

## Supporting information

**S1 Fig. Plant cultivation. (A)** Sieved and homogenized field soil ready for distribution into pots for experiments in Tübingen. **(B)** Field-grown plants in Tübingen under snowfall in January. **(C)** Field-grown plants in Tübingen from 2016 to 2017 season on the day of harvest. **(D)** Greenhouse-grown plants in Tübingen from 2016 to 2017 season. **(E)** Greenhouse-grown plants in Tübingen from 2016 to 2017 season on the day of harvest. **(F)** Field-grown plants in Zurich from 2018 to 2019. **(G)** Closeup of a chain-link fence as an open side partition at the open-air room at the Zurich field site. The ceiling was open except for a net. Wind, rain, and insects could freely pass through. **(H)** Images of field plants from Tübingen on day or harvest before (left) and after (right) background removal. **(I)** Images of greenhouse plants from Tübingen on day or harvest before (left) and after (right) background removal. Note that overlap of leaves from plants in adjacent pots occasionally required manual estimation of leaf borders. (TIF)

**S2 Fig. Expression of various immunity marker genes.** Normalized RNA-seq read counts (y-axes) for different genes with relevance to immune system activity. Colors of boxplots indicate plant genotype as described at the top left. The colored annotations along the x-axis denote the environment (GH-Tü = Greenhouse Tübingen, F-Tü = Field Tübingen,

F-Z = Field Zurich). * signifies $P < 0.01$ in an FDR-corrected Wald test. The data underlying this figure can be found in https://doi.org/10.5281/zenodo.15527338.
(TIF)

**S3 Fig. Comparison of differential expression based on magnitude of *P*-value or expression change. (A)** Overlaps between upregulated (black numbers) and downregulated (red numbers) genes in *ACD6* allelic contrasts, either in the greenhouse (top) or field (bottom), for the 100 genes with the lowest *P* values from each contrast (regardless of any significance threshold). **(B)** Similar to A, but for the 100 genes from each contrast with the largest fold change (regardless of any significance threshold). The data underlying this figure can be found in https://doi.org/10.5281/zenodo.15527338.
(TIF)

**S4 Fig. Gene ontology enrichments for the 100 upregulated genes with the lowest *P*-values in Est-1 or HIF:Est-1 in all environments.** The top 100 upregulated genes are defined as those with the lowest *P*-values, regardless of a *P*-value threshold. In both greenhouse comparisons, multiple immune system processes are significantly enriched among these 100 genes. In both field comparisons, no biological process is significantly enriched among these 100 genes. Obs., bserved count; Exp., Expected count; FE, Fold enrichment; Adj. *P* is the FDR-corrected *P*-value.
(TIF)

**S5 Fig. Phytohormone measurements.** From rosettes grown in Tübingen, salicylic acid (SA), its inactivate storage form SA O-β-glucoside (SAG), jasmonic acid (JA), indole-3-acetic acid (IAA), and indole-3-carboxylic acid (ICA) were measured from snap-frozen tissue via LC-MS. Letters "a" and "b" above the boxplots represent groups that are statistically different in a FDR-corrected Mann–Whitney *U*-test ($P < 0.05$) across three comparisons: HIF:Col-0 vs. HIF:Est-1 in the greenhouse, Est-1:*acd6*-null vs. Est-1 in the greenhouse, and Est-1 in the greenhouse vs. Est-1 in the field. The data underlying this figure can be found in https://doi.org/10.5281/zenodo.15527338.
(TIF)

**S6 Fig. *ACD6*-dependent growth differences at 16 and 23°C.** Twenty-four Est-1 and 24 Est-1:*ACD6*-null seeds were sown in potting soil, grown in either 16 or 23°C long (16 h) days, and green pixels were monitored daily from overhead photographs starting 1 week after sowing. In both temperature regimes. The data underlying this figure can be found in https://doi.org/10.5281/zenodo.15527338.
(TIF)

**S7 Fig. Artificial infections reveal subtle *ACD6*-dependent differences in bacterial assembly in the greenhouse. (A)** Bacterial loads of the most abundant ASVs in field-grown HIF plants challenged with a cocktail of PvATUE5 strains in the field for 1 week, with color conventions for the plant genotypes as shown in the legend. The ASV corresponding to the cocktail of PvATUE5 is labeled in red. Boxes enclose the interquartile range (IQR) with whiskers extending to up to 1.5 times the IQR. **(B)** Overall bacterial load considering all ASVs for the plants in (A). **(C)** Overall bacterial load for all ASVs for growth chamber-grown Est-1 and Est-1:*acd6*-null plants challenged with Pst DC3000 and/or other phyllosphere bacteria for 4 days, with color code as in (A). Red points indicate samples with especially high bacterial load for which hamPCR could not provide accurate quantification due to a small number of reads from the plant's GIGANTEA gene. d, Colony forming units (CFUs) of Pst DC3000 per $cm^2$ of leaf tissue for Pst DC3000-only infected plants shown in (C). * indicates $P < 0.05$ in a Mann–Whitney *U*-test. **(E)** Bacterial loads of the most abundant ASVs in growth chamber-grown plants infected with Pst DC3000 only as shown in (C) and (D). Red points indicate samples with especially high bacterial load for which hamPCR could not provide accurate quantification due to a small number of reads from the plant's GIGANTEA gene. Sample size without considering the red points is too small and statistics are, therefore, omitted. The ASV

corresponding to Pst DC3000 is labeled in red. The data underlying this figure can be found in https://doi.org/10.5281/zenodo.15527338.

(TIF)

**S8 Fig. Sample numbers for each condition in various experiments.**
(TIF)

## Acknowledgments

We thank Roosa Laitinen (University of Helsinki), Kyle Gervers (Swedish University of Agricultural Sciences), and Sur Herrera Paredes (National Autonomous University of Mexico) for helpful comments on the manuscript and Eric Kemen (University of Tübingen) for helpful comments on experimental approaches. We thank Matthew Horton and Jana Mittelstrass (University of Zürich) for helping to arrange planting facilities at Agroscope (Reckenholz) in Switzerland.

The "harvest team" comprises all the above authors plus: Adrián Contreras, Anjar Tri Wibowo, Bridgit Waithaka, Cristina Barragan, Efthymia Symeonidi, Frank Vogt, Hua Wang, Julia Elis, Lei Li, Moises Exposito Alonso, Nilufar Yesmin, Or Shalev, Patricia Lang, Rui Wu, Sergio Latorre, Talia Karasov, Thanvi Srikant, and Ulrich Lutz.

## Author contributions

**Conceptualization:** Derek S. Lundberg, Detlef Weigel.

**Data curation:** Derek S. Lundberg, Ezgi Mehmetoğlu Boz, Pratchaya Pramoj Na Ayutthaya.

**Formal analysis:** Derek S. Lundberg.

**Funding acquisition:** Derek S. Lundberg, Detlef Weigel.

**Investigation:** Derek S. Lundberg, Sonja Kersten, Ezgi Mehmetoğlu Boz, Pratchaya Pramoj Na Ayutthaya, Karin Poersch, Sophia Swartz, David Müller, Detlef Weigel.

**Methodology:** Derek S. Lundberg, Sonja Kersten, Wangsheng Zhu.

**Resources:** Wangsheng Zhu, Wei Yuan, Detlef Weigel.

**Software:** Ilja Bezrukov.

**Supervision:** Derek S. Lundberg, Detlef Weigel.

**Visualization:** Derek S. Lundberg, Ezgi Mehmetoğlu Boz, Detlef Weigel.

**Writing—original draft:** Derek S. Lundberg, Pratchaya Pramoj Na Ayutthaya, Detlef Weigel.

**Writing—review & editing:** Derek S. Lundberg, Ezgi Mehmetoğlu Boz, Pratchaya Pramoj Na Ayutthaya, Wangsheng Zhu, Detlef Weigel.

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
