## [Editor Report · Decision Letter 0]

Dear Dr Lundberg,

Thank you for submitting your manuscript entitled "Field conditions greatly modify a major growth-defense tradeoff in Arabidopsis thaliana" for consideration as a Research Article by PLOS Biology.

Your manuscript has now been evaluated by the PLOS Biology editorial staff, as well as by an academic editor with relevant expertise, and I'm writing to let you know that we would like to send your submission out for external peer review.

Once your full submission is complete, your paper will undergo a series of checks in preparation for peer review. After your manuscript has passed the checks it will be sent out for review. To provide the metadata for your submission, please Login to Editorial Manager (https://www.editorialmanager.com/pbiology) within two working days, i.e. by Oct 31 2024 11:59PM.

Kind regards,

Roli Roberts

Roland Roberts, PhD

Senior Editor

PLOS Biology

rroberts@plos.org

---

## [Decision Letter · Decision Letter 1]

Dear Dr Lundberg,

Thank you for your patience while your manuscript "Field conditions greatly modify a major growth-defense tradeoff in Arabidopsis thaliana" went through peer-review at PLOS Biology. Your manuscript has now been evaluated by the PLOS Biology editors, an Academic Editor with relevant expertise, and by two independent reviewers. Please accept my apologies for the delay incurred over the holiday period.

You'll see that reviewer #1 is very positive and has no requests. Reviewer #2 is also largely positive, but wants much clearer description of the methods, especially the stats, and thinks that the framing of the Intro and Discussion needs to be improved in order to bring out the advance for a broader audience (their point 3).

IMPORTANT: I discussed these comments with the Academic Editor, and we both agree fully with the points raised by reviewer #2, and especially the need to bring out the advance that it most relevant to the broader readership, namely the striking and reproducible disparity between field and greenhouse experiments.

In light of the reviews, which you will find at the end of this email, we are pleased to offer you the opportunity to address the [comments/remaining points] from the reviewers in a revision that we anticipate should not take you very long. We will then assess your revised manuscript and your response to the reviewers' comments with our Academic Editor aiming to avoid further rounds of peer-review, although might need to consult with the reviewers, depending on the nature of the revisions.

**IMPORTANT - SUBMITTING YOUR REVISION**

*Resubmission Checklist*

*Published Peer Review*

*PLOS Data Policy*

*Blot and Gel Data Policy*

Sincerely,

Roli Roberts

Roland Roberts, PhD

Senior Editor

PLOS Biology

rroberts@plos.org

REVIEWERS' COMMENTS:

Reviewer #1:

In this paper, Lundberg et al. investigate effects of ACD6 alleles, using parallel approaches with near-isogenic lines and gene editing. ACD6 is associated with defense responses, which typically come at the cost of reduced growth and fecundity. The authors studied these growth-defense tradeoffs in a series of field and greenhouse experiments, in great detail focusing on visual phenotypic traits, gene expression, hormone levels, and leaf microbiome composition.

The key finding, which really is surprising and extremely important, is that the expected growth-defense tradeoff only manifested under greenhouse conditions. There, defense-associated gene expression was high in the corresponding genotypes, and associated costs such as reduced growth and leaf necrosis were observed. In the field, these effects were not observed, neither in near-isogenic lines derived from Col-0 x Est-1 crosses, nor in comparisons of lines in which the Est-1 allele was knocked out by editing with CRISPR/Cas9.

Overall, these strong G x E interaction demonstrate that findings from controlled-condition studies may not be transferable to the field, which has far-reaching consequences. One would have to understand the specific environmental drivers of this GxE interaction, which the authors have tried. They investigated and discuss different possibilities, including differences in temperature between field and greenhouse and differences in foliar microbiome, but did not find an explanation. I would like to stress that the authors are not to blame for this lack of explanation. On the contrary, I highly appreciate the level of detail in which this was done, and how transparently this field-greenhouse discrepancy is presented.

This fascinating paper was a pleasure to read, is well written, the methods clearly described, and in my opinion publishable as is.

Reviewer #2:

Overall this is a high quality manuscript describing an interesting partial-null result, in which naturally occurring allelic variants of the ACD6 gene in Arabidopsis thaliana have major effects on growth and disease resistance in greenhouse experiments, but not in the outdoors. It is well written and (except for the statistical analysis, see detailed comments below) the work appears to be well executed. The authors demonstrate this using both heterogeneous inbred families to compare the two alleles directly to each other, and using a mutant line to compare one of the two alleles to a knockout. The HIF approach is more useful for understanding the functional consequences of naturally occurring genetic variation, but it is nice that the results from the two methods made for a consistent story. I appreciated that the alleles' effects (or lack thereof) were assessed both at the phenotype level and the molecular (transcription) level, and also that the authors tested for evidence of selection in the invisible fraction by bulk-genotyping multiple generations of plants in each environment. These elements make the paper a fairly thorough exploration of the fitness consequences of ACD6 in greenhouse and field environments, with a particular focus on its effect on plant-microbe interactions.

Here are my major comments.

1. Unfortunately, I am unable to evaluate the data analysis because many key details were missing from the manuscript. Basic information such as the sample sizes for each experiment, replication of each genotype, was not provided. For the DESeq2 analysis, the model being tested was not mentioned, making it impossible to interpret the results. For example, did the model simultaneously account for genotype and environment or was it done separately for each environment? Details about the statistical analysis of rosette size data were not provided - the figure captions mention Mann-Whitney U tests, which implies that many such tests were conducted to compare pairs of genotypes in various experiments and environments. This is a classic case where P-values must be adjusted to correct for the inflated Type I error rate, but there is no mention of this being done.

2. Following my first comment: Anyway, repeated pairwise tests cannot provide sufficient statistical support for the main conclusion of this paper. As aptly put by Gelman and Stern in 2012 (https://doi.org/10.1198/000313006X152649), the difference between P<0.05 in one environment and P>0.05 in a second environment is not necessarily statistically significant. Fundamentally this paper is about a genotype-by-environment interaction, which corresponds to a hypothesis that can only be tested with an interaction term followed by post-hoc testing. This applies to the rosette size data as well as the gene expression and microbiome analyses.

3. In my opinion the potential impact is unclear and the paper is missing a broader context. Will these results be interesting to readers who are not interested in Arabidopsis genetics, plant immunology, or microbiomes? I do see how it is useful as a cautionary tale for inferring ecological and evolutionary phenomena from greenhouse studies. The authors do mention balancing selection but it is not really clear how this story advances our understanding of balancing selection. Possibly this just requires some re-framing in the introduction and discussion.

4. I am confused about how and when the HIF individuals were genotyped - page 4 states that only the HIF:Est-1 and HIF:Col-0 genotypes (along with Est-1 wildtype) were planted in the greenhouse and outdoors, but the method for genotyping required destructive harvesting. Presumably half of the HIF seed would have been heterozygous, were these also included in the experiment but their phenotypic data not analyzed? Or was there some way of discerning ACD6 genotype prior to planting in the field?

5. Figure 2c, it was not clear whether this analysis used data from all environments, or one environment. The methods for this analysis seem to be missing as well.

6. More information on the "secure open-air room" field setting would be helpful to put in context how comparable the results should be to a "normal" field setting. For example, how is it enclosed? Is it accessible to insects? Are the plants exposed to wind?

7. A few unexplained acronyms - page 12, Pv-ATUE5, I assume these are different species of Pseudomonas? Page 3, EMS

8. Figure 4, panel c.i, the "NS" arrows seem to be pointing to nothing

---

## [Decision Letter · Decision Letter 2]

Dear Dr Lundberg,

Thank you for your patience while we considered your revised manuscript "Field conditions greatly modify a major growth-defense tradeoff in Arabidopsis thaliana" for publication as a Short Report at PLOS Biology. This revised version of your manuscript has been evaluated by the PLOS Biology editors, the Academic Editor and one of the original reviewers.

Based on the review, we are likely to accept this manuscript for publication, provided you satisfactorily address the following data and other policy-related requests.

IMPORTANT - please attend to the following:

a) Please could you make your Title more explicit and accessible to a broader readership? We suggest: "A major tradeoff between growth and pathogen defense in hyperactive ACD6 Arabidopsis thaliana is absent in field conditions"

b) After some discussion, we've decided that your paper would be more appropriate for a Short Report format. As your manuscript is already succinct, no re-formatting is needed, and I have simply flipped the article type on your behalf. No further action is required.

c) Please address my Data Policy requests below; specifically, we need you to supply the numerical values underlying Figs 1B, 2ABDEFG, 3C, 4ABCD, S2, S3, S5, S6AB, S7ABCDE, either as a supplementary data file or as a permanent DOI’d deposition.

d) Please cite the location of the data clearly in all relevant main and supplementary Figure legends, e.g. “The data underlying this Figure can be found in S1 Data” or “The data underlying this Figure can be found in https://zenodo.org/records/XXXXXXXX

e) Please make any custom code available, either as a supplementary file or as part of your data deposition.

We expect to receive your revised manuscript within two weeks.

*Published Peer Review History*

*Press*

Sincerely,

Roli Roberts

Roland Roberts, PhD

Senior Editor

rroberts@plos.org

PLOS Biology

DATA POLICY:

Regardless of the method selected, please ensure that you provide the individual numerical values that underlie the summary data displayed in the following figure panels as they are essential for readers to assess your analysis and to reproduce it: Figs 1B, 2ABDEFG, 3C, 4ABCD, S2, S3, S5, S6AB, S7ABCDE. NOTE: the numerical data provided should include all replicates AND the way in which the plotted mean and errors were derived (it should not present only the mean/average values).

CODE POLICY

DATA NOT SHOWN?

REVIEWER'S COMMENTS:

Reviewer #2:

The authors have done a good job of addressing my concerns with the original manuscript, particularly by presenting their results focused on effect sizes, and by adding statistical detail. I also thought the introduction was much improved, and the paper is an interesting story of a polymorphism that appears to be maintained by balancing selection, but which resists characterization under natural field conditions.

---

## [Editor Report · Decision Letter 3]

Dear Dr Lundberg,

Thank you for the submission of your revised Short Report "A major tradeoff between growth and defense in Arabidopsis thaliana can vanish in field conditions" for publication in PLOS Biology. On behalf of my colleagues and the Academic Editor, Pamela Ronald, I'm pleased to say that we can in principle accept your manuscript for publication, provided you address any remaining formatting and reporting issues. These will be detailed in an email you should receive within 2-3 business days from our colleagues in the journal operations team; no action is required from you until then. Please note that we will not be able to formally accept your manuscript and schedule it for publication until you have completed any requested changes.

Sincerely, 

Roli Roberts

Senior Editor

PLOS Biology

rroberts@plos.org